# Depth of Remission Following First-Line Treatment Is an Independent Prognostic Marker for Progression-Free Survival in Gastric Mucosa-Associated Lymphoid Tissue (MALT) Lymphoma

**DOI:** 10.3390/cancers12020492

**Published:** 2020-02-20

**Authors:** Barbara Kiesewetter, Ingrid Simonitsch-Klupp, Werner Dolak, Marius E. Mayerhoefer, Markus Raderer

**Affiliations:** 1Department of Medicine I, Clinical Division of Oncology, Medical University of Vienna, A-1090 Vienna, Austria; barbara.kiesewetter@meduniwien.ac.at; 2Department of Pathology, Medical University of Vienna, A-1090 Vienna, Austria; ingrid.simonitsch-klupp@meduniwien.ac.at; 3Department of Medicine III, Clinical Division of Gastroenterology and Hepatology, Medical University of Vienna, A-1090 Vienna, Austria; werner.dolak@meduniwien.ac.at; 4Department of Biomedical Imaging and Image-guided Therapy, Division of Nuclear Medicine, Medical University of Vienna, A-1090 Vienna, Austria; marius.mayerhoefer@meduniwien.ac.at

**Keywords:** MALT lymphoma, gastric MALT lymphoma, indolent lymphoma, extranodal lymphoma, *Helicobacter pylori*, prognostic factors, treatment

## Abstract

Gastric mucosa-associated lymphoid tissue (MALT) lymphoma responding to upfront treatment has an excellent outcome and no further therapy is recommended, even in the presence of residual disease. However, no data exist on the influence of initial depth of remission on progression-free survival (PFS). *Methods*: We investigated a correlation between PFS and depth of response, categorizing them as complete remission (CR), partial remission (PR) and stable disease (SD) in 137 consecutive patients at the Medical University Vienna. *Results*: All patients with *Helicobacter pylori* (*H. pylori*)-positive, localized disease received *H. pylori* eradication (70%, 96/137), while the remaining patients were treated with various modalities. The response rate was 67% for the entire collective and 58% for eradication only, with corresponding CR-rates of 48% and 38%. At a median follow-up of 56.2 months, the estimated PFS for the entire cohort was 34.2 months (95% Confidence Interval 16.0–52.4). Responding patients (=CR/PR) had a significantly longer PFS compared to SD (68.3 vs. 17.3 months, *p* < 0.001). This was also applicable to the eradication only cohort (49.0 vs. 17.3 months, *p* < 0.001) and remained significant after correction for MALT-IPI. Furthermore, CR significantly prolonged PFS over PR (*p* = 0.007 entire cohort, *p* = 0.020 eradication). *Conclusions:* Remission status correlated significantly with PFS, suggesting depth of remission as prognostic marker for long-term relapse-free survival.

## 1. Introduction

Mucosa-associated lymphoid tissue lymphoma (MALT lymphoma) represents one of the more common subtypes of indolent lymphoma and occurs at an estimated age-adjusted incidence rate of 1.1/100.000 [1]. Deriving from mature B-cells and histologically characterized by the variable presence of lymphoepithelial lesions, follicular colonization and plasmacellular differentiation, this distinct lymphoma entity has the ability to develop into acquired mucosa-associated lymphoid structures throughout the entire body [2]. While literally every organ may be involved, the gastric mucosa is the most common localization, accounting for up to 50% of all MALT lymphomas and representing also the most frequent type of primary gastric lymphoma [3]. In contrast to other B-cell lymphomas, MALT lymphoma is a disease driven by immunological factors, with chronic infections and autoimmune processes constituting the most relevant defined triggers [4]. Studies from the 1990s have confirmed the association between chronic *Helicobacter pylori (H. pylori)* gastritis and MALT lymphomagenesis, which is considered to be a multi-stage process based on the accrual of *H. pylori* specific T-cells and perpetual antigenic stimulation of marginal zone B-cells [5]. Thus, the antibiotic eradication of *H. pylori* constitutes the current standard of care for upfront management of such patients [6,7].

Long-term outcome following first line antibiotic eradication for localized gastric MALT lymphoma is favorable with up to 80% durable remissions and 5-year survival rates of approximately 90% according to most reported series [8,9,10,11,12]. However, while the rate of complete remissions (CR) is reported to be in the range of 60–75%, approximately one third of patients show persistent disease in follow-up biopsies classified either as no change (NC), or with signs of lymphoma regression described as responding residual disease (rRD) or probable minimal residual disease (pMRD), according to GELA (Groupe d’ Etude des Lymphomes de l’ Adulte) histological response criteria, a grading system specifically developed for post-treatment biopsies of gastric MALT lymphoma [13,14]. In order to avoid “over-treatment” in this indolent lymphoma entity and given the observation that patients with residual disease may safely be followed without increased risk for transformation, the current guidelines recommend that in the absence of clinical and/or histological progression, no further treatment is indicated after *H. pylori* eradication and that patients with persisting lymphoma infiltration but no sign of progression may be followed by regular endoscopic examinations and clinical controls [6,7,15]. Furthermore, the time to response to *H. pylori* eradication is known to be variable and objective responses may be observed even after a prolonged period of time.

However, no data on long-term outcome of patients with persisting or residual disease in terms of relapse patterns and progression-free survival (PFS) have been published to judge whether these are influenced by the remission status and particularly the “depth” of remission, i.e., CR versus responding disease (rRD, mRD) and no change (NC) cases. As this might influence further clinical management of patients following *H. pylori* eradication and might also potentially allow the development of more stratified endoscopic follow-up strategies—which, in fact, still constitute a physical burden to the patient. We systematically investigated 137 consecutive patients with gastric MALT lymphoma treated at the Medical University of Vienna, a tertiary referral center for extranodal lymphoma, for response to first line therapy and subsequent relapse patterns with a particular focus on patients treated upfront with *H. pylori* eradication.

## 2. Methods

For this analysis, we retrospectively evaluated all patients treated for histologically verified MALT lymphoma between 1999 and 2019 at the Clinical Division of Oncology of the Medical University of Vienna. Only patients classified as having primary gastric MALT lymphoma were included. All diagnoses were established by a reference pathologist according to the most recent World Health Organization (WHO) Classification of Tumors of Hematopoietic and Lymphoid Tissues including adequate immunophenotyping on paraffin-embedded specimen, i.e., CD20+CD5-CD10-cyclinD1- and demonstration of light chain restriction [16]. Furthermore, the presence/absence of plasmacytic differentiation was documented. Patients and corresponding clinical data were extracted from electronic and/or paper-based medical records routinely collected and stored at the department. This analysis was approved by the local ethical committee of the Medical University of Vienna (EK No. 791/2011, 13 September 2019).

### 2.1. Evaluated Data

Data extracted for this analysis included basic patient characteristics i.e., sex, age at initial diagnosis, performance status, localization of primary disease, stage of disease according to the Ann Arbor classification and the Lugano staging system, lymphoma-related laboratory findings, including lactate dehydrogenase (LDH) levels, beta-2-microglobulin (B2M) levels and the presence of paraproteinemia at first presentation, and furthermore, *H. pylori* status both on histological samples as well as assessed by IgG-specific serology and Hepatitis virus B/C status. In addition, all MALT lymphoma patients were routinely screened for autoimmune disorders and results were documented. The MALT-IPI, composed of number of extranodal sites, LDH and age 70+, was retrospectively calculated for risk stratification if possible [17]. Regarding first-line treatment, we collected type of treatment applied and corresponding response/outcome. Subsequent relapses, progression-free survival (PFS) and time to next treatment (TTNT) were documented according to the revised response criteria for malignant lymphoma [18]. Finally, follow-up time and overall survival (OS) were assessed, including cause of death if applicable.

### 2.2. Staging, Response Assessment and Follow-Up

All patients at our department were staged and followed according to a standardized protocol as published before [19], allowing an unbiased longitudinal documentation of follow-up. In addition to endoscopic assessment, including gastro-duodenoscopy, endosonography of the upper-gastrointestinal (GI-)tract and colonoscopy with multiple biopsies, a computer tomography (CT) scan of thorax and abdomen or a comparable systemic staging modality was performed at least once at initial diagnosis to identify potential extragastric disease. Endoscopic controls with biopsies as recommended by the current guidelines were performed every three months in the first two years and every six months afterwards. In case of radiologically measurable disease, systemic imaging was also repeated on the same schedule. Response was (re-)classified based on histology as defined per GELA histological response criteria [13,14], i.e., CR, pMRD, rRD, NC, and in case of radiologically measurable disease, in addition, per radiological response criteria, i.e., CR, partial remission (PR), stable disease (SD) and progressive disease (PD). According to current clinical guidelines for gastric MALT lymphoma, 18F-FDG-PET/CT was not routinely used for assessment of the disease. For all response-related analyses in this study, rRD and pMRD were summarized as partial remission (PR) and no change (NC) was considered as stable disease (SD).

### 2.3. Statistical Analysis

A statistical analysis was performed using IBM Statistics for Mac OS version 26.0 (IBM, Armonk, NY, USA). Metric data were described using median, range and interquartile range (IQR). In case of categorical data, we present percentages and absolute frequencies. Associations of binary variables were assessed by use of the Chi-square test. Estimated PFS and OS were plotted by the Kaplan–Meier method and differences between groups were compared using log rank testing. A proportional cox-regression model was used to integrate MALT-IPI factors into PFS analyses. The Student’s *t*-test was used to compare means between two groups. *p* values < 0.05 (two-sided) were considered statistically significant.

## 3. Results

### 3.1. Basic Characteristics 

We identified 412 patients with MALT lymphoma diagnosed and treated at our department from 1999 to 2019, including a total of 137 patients (33%, 137/412) classified as primary gastric MALT lymphoma. In this group, the ratio of female to male patients was nearly balanced at 1.1 (51% female, 49% male) and the median age at initial diagnosis of lymphoma was 63 years (range, 22–85; interquartile range, 52–72). Stage of disease according to Ann Arbor was localized in the majority of cases with 66% confined to the stomach (= IE Ann Arbor), 23% presenting with local lymph node involvement (= IIE Ann Arbor), one patient presenting with distant lymph node involvement (1%, = IIIE Ann Arbor) and only 10% diagnosed with disseminated disease (=IV Ann Arbor); for classification according to the Lugano staging system, see Table 1. Other organs involved in patients with stage IV disease included the lung (*n* = 4), the colon (*n* = 4), the ocular adnexa (*n* = 2), the bone marrow (*n* = 2), the spleen (*n* = 2), the bladder (*n* = 1), the tonsils (*n* = 1), the kidneys (*n* = 1), the peritoneum (*n* = 1), and the parotid glands (*n* = 1). As expected, the performance status was excellent in most patients and 85% presented with a performance status of 0, 12% with a performance status of 1 and only 2% of patients each with a performance status of 2 and 3. *H. pylori* status as assessed on tissues and/or serology was found to be positive in 68% of patients, while 32% of patients did not show any evidence of infection. In 89% of patients (122/137), all requested factors for calculation of the MALT-IPI were available, categorizing 62% of patients as low risk, 33% as intermediate risk and 6% as high risk in terms of long-term outcome (for more detailed characteristics, see Table 1).

### 3.2. First-line treatment and Response 

According to the current guidelines, all patients with localized H. pylori-positive gastric MALT lymphoma received adequate eradication treatment. Patients with H. pylori-negative MALT lymphoma were initially not treated with *H. pylori* eradication but were subjected to antibiotics only in more recent years for upfront management, resulting in 22/44 patients with negative *H. pylori* status being only treated with antibiotics (see Table 2). The remaining 22 H. pylori-negative patients and those with primary disseminated disease were treated with various therapeutic strategies, including local treatment options (radiotherapy, surgery) or systemic therapy with chemo- and/or immunotherapy. Consequently, in total, 70% (96/137) of patients received upfront sole antibiotic eradication while 23% (32/137) of patients were treated with systemic therapy and a small group of 4% (5/137) with a local treatment modality. The final 3% (4/137) of patients did not receive any treatment and were handled with a watch and wait strategy only; these patients were excluded from the objective response assessment.

In terms of objective response to first-line treatment, data were available in 96% (128/133) of actively treated patients. The documented overall response rate for the entire collective was 67% (86/128), the rate of histological and radiological CR was 48% (62/128), the PR rate 19% (24/128), and 30% (38/128) showed disease stabilization as the best response while only 3% (4/128) had primary progressive disease. In the *H. pylori* eradication-only cohort, an objective response was documented in 58% (54/93), a CR in 38% (35/93), a PR in 20% (19/93), and stabilization of disease as best response in 39% (36/93). As expected, the overall response rate was higher in H. pylori-positive patients (62%, 44/71) than in H. pylori-negative patients (46%, 10/22). The rate of primary PD in the antibiotic-only cohort was identical to the entire cohort at 3% (3/93). For a detailed response to systemic treatment and local therapy, see Table 2.

Time to response was documented in all patients achieving an objective response, i.e., PR or CR, and was significantly longer in patients with *H. pylori* eradication than in patients treated with either systemic or local therapy up-front (mean 7.6 months versus 4.3 months, *p* = 0.002). As expected, the time to response following *H. pylori* eradication was highly variable, with the longest time to best response documented being 32.2 months in this collective.

### 3.3. Progression-Free Survival

At a median follow-up time of 56.2 months (IQR 26.4–111.6), a total of 52% of patients (69/132, lost to follow-up *n* = 5) had relapsed or progressed. The median-estimated PFS for the entire collective was 34.2 months (95%CI 16.0–52.4) (Figure 1). Remarkably, estimated PFS was significantly longer in patients achieving an objective response to treatment, i.e., CR or PR, with 68.3 months versus 17.3 months in patients with SD/NC as best response to treatment (*p* < 0.001) (Figure 2). Furthermore, patients achieving a CR had a significantly longer estimated median PFS of 94.8 months versus 28.5 months in patients showing only a PR (*p* = 0.007) (Figure 3). These results remained statistically significant after multivariate correction for MALT-IPI factors (*p* < 0.001 and *p* = 0.031, respectively). No correlation between PFS and diagnosis of autoimmune disorder, plasmacellular differentiation or the presence of paraproteinemia was observed (*p*-values all non-significant).

In terms of further treatment, 49% of patients received subsequent second line therapy, which was systemic therapy with immuno-/ chemotherapy in the majority of cases (85%; 8% local therapy, 8% antibiotics). Patients with an initial objective response were significantly less likely to receive second-line treatment compared to patients with SD as the best response (41% versus 63%, *p* = 0.021), while in CR versus PR there was only a non-significant trend detected (39% versus 46%, *p* = 0.546). In patients undergoing second-line treatment, time to next treatment (TTNT) was significantly longer for responders versus SD (mean 47.5 months versus 15.3 months, *p* = 0.001), whereas in CR versus PR, there was only a slight numerical difference (mean 52.7 months versus 35.1 months, *p* = 0.385).

Looking specifically at the *H. pylori* eradication cohort (*n* = 96), a total of 49% of patients (46/94, lost to follow-up *n* = 2) relapsed after first-line treatment and the median estimated PFS for this group was 27.6 months (95% CI 22.6–32.6). Again, PFS was significantly longer for patients who achieved an objective response (CR/PR) with an estimated median PFS of 49.0 months versus 17.3 months in patients with SD/NC as best outcome (*p* < 0.001) (Figure 4). Furthermore, CR also increased PFS significantly over PR with 109.4 versus 27.6 months (*p* = 0.020) (Figure 5). These results remained significant after multivariate correction for MALT-IPI factors in case of responders versus SD (*p* < 0.001) and only marginally failed statistical significance in the group of CR versus PR (*p* = 0.058). No correlation between PFS and diagnosis of autoimmune disorder, plasmacellular differentiation or the presence of paraproteinemia was observed (*p*-values all non-significant).

In the *H. pylori* eradication cohort, 46% (44/96) of patients received subsequent second-line therapy, with the large majority being treated with immuno-/chemotherapy (89%; 7% antibiotics, 5% local treatment). In line with results for the entire cohort, patients with an objective response to first-line treatment had a significantly lower likelihood of receiving second-line treatment compared to patients with SD as the best response (33% versus 61%, *p* = 0.009), while in CR versus PR, there was only a non-significant trend detected (29% versus 42%, *p* = 0.314). In patients receiving second-line treatment, TTNT was significantly longer for responders versus non-responders (mean 30.3 months versus 15.4 months, *p* = 0.008), but TTNT did only tendentially differ for CR versus PR (mean 34.2 months versus 25.5 months, *p* = 0.439). Notably, in the *H. pylori* up-front eradication group, three patients had a H. pylori-positive gastric relapse and received re-eradication. Of these, one again showed an objective response (CR) while one had persistent (GELA no change) MALT lymphoma on endoscopic controls and one progressed shortly after and was subsequently salvaged with systemic treatment.

### 3.4. Long-Term Outcome and Overall Survival

As expected, overall survival at 5 and 10 years was excellent, with 91% and 79%, respectively, and only four patients (3%, 4/137) died related to B-cell lymphoma. However, in contrast to the prognostic value of depth of remission for PFS, overall survival did not differ between responders and CR/PR for any cohort (*p*-values non-significant). Cause of lymphoma-related death was progression of MALT lymphoma 302 months after initial diagnosis in one patient, gastric bleeding in one patient, progression of an unrelated EBV-associated B-cell lymphoma in one patient, and transformation to diffuse large B-cell lymphoma (DLBCL) in the last patient (DLBCL clonally related, transformation 22 months after diagnosis). The final transformation rate in our collective of gastric MALT lymphomas was low (6/137; 4%); apart from the single patient who died, the remaining 5 patients achieved a CR with R-CHOP therapy underscoring the good prognosis of gastric MALT lymphoma, even in case of transformation as opposed to extragastric MALT lymphomas developing DLBCL [20].

## 4. Discussion

Gastric MALT lymphoma is an indolent disease, and our data underscore the excellent outcome and overall survival of these patients in this single-center series with standardized and prolonged follow-up. In our total cohort of 412 consecutive patients with MALT lymphoma, 137 (33%) were primary gastric, which is lower than in the older series but in keeping with the recent trend reporting a predominance of extragastric MALT lymphomas [3]. Also in line with the current literature are the characteristics of gender, age, stage and MALT-IPI, with the latter being low in the large majority of patients while only 6% had a high risk MALT-IPI (MALT-IPI score > 1). In addition, the relatively high rate of H. pylori-negative patients in our series is in keeping with recent data and the overall cohort appears representative in terms of clinical characteristics if compared to other centers.

The indolent nature of the disease in the case of lymphoma-persistence after antibiotic therapy [8] and the fact that chlorambucil did not result in a better outcome following *H. pylori* eradication over wait-and-see in all subgroups [21] have led to the recommendation that patients responding to first line antibiotics should not undergo further therapy in order to “force” a complete remission [6,7]. In addition, relatively scarce results on follow-up and clinical course after frontline systemic therapy in gastric MALT lymphoma have been published, so no such data and guidelines exist in the current literature. Due to the widespread use of radiotherapy in some countries, only the relapse rate in localized gastric MALT lymphoma can be assessed from recent publications but no distinction between responders and stable disease or furthermore CR versus PR has been made, which is most likely due to the high local activity of the radiation schedules applied resulting in virtually all patients achieving a CR [22].

In our series, the median follow-up was 56.2 months, which appears sufficiently long to reliably judge the clinical course of our collective. In fact, this is comparable to other larger series and is well within or exceeding the range of the median time to relapse/progression, which is between 4–5 years in the current literature [3,23]. In addition, a prolonged follow-up to correctly assess response is especially important in patients undergoing *H. pylori* eradication, where individual time to best response can be in excess of two years [3,6,7,8,9,10,11,12]. This was also highlighted in a study by Fischbach et al. in which 32% of patients in PR after 12 months consecutively converted to CR with prolonged follow-up (median follow-up of 42 months) [15].

In the current cohort of 137 patients with gastric lymphoma, PFS following first-line treatment was significantly longer in responders (=PR/CR) versus patients who only achieved stable disease (*p* < 0.001), with the estimated median PFS being 68.3 months versus 17.3 months for these groups. Furthermore, patients achieving a CR had a significantly longer PFS of 94.8 months as opposed to 28.5 months in patients developing a PR as best response (*p* = 0.007). These results remained statistically significant after correction for MALT-IPI factors (*p* < 0.001 and *p* = 0.031, respectively). Importantly, the significant impact of response to initial therapy was also apparent in the largest subgroup of patients with *H. pylori* eradication, with an estimated median PFS of 49.0 months for PR/CR versus 17.3 months in patients with SD (*p* < 0.001). Again, CR was also associated with a significantly increased PFS over PR (*p* = 0.020).

These results show that the risk of relapse/progression markedly differs not only between patients in CR versus remaining lymphoma but also between patients in PR (in our series summarizing rRD and pMRD) and stable disease. While the current guidelines already differentiate between CR and remaining lymphoma in the recommendation for endoscopic follow-up (i.e., every 6 months for two years, then every 12–18 months for CR, but every 3–6 months for patients with lymphoma-remnants) [7], our data suggest that patients with response to initial antibiotics might also undergo a less strict endoscopic follow-up routine in the absence of symptoms suspicious of worsening of the disease. While our data do not allow for an exact recommendation, a follow-up interval of 6–12 months for responders appears sensible but should probably be studied prospectively. However, the need for a prolonged assessment of such patients as proposed by earlier trials is again highlighted in our study in view of the delayed potential for relapse following initial therapy [3,6,7,8,24].

In addition to PFS time to next treatment (TTNT) was also significantly different between responders versus non-responders both in the whole cohort as well as the *H. pylori* eradication-only group. This, however, may be a partially subjective parameter, based on patient preference, symptoms and the personal perception of the treating physician judging a potentially responsive disease as less dangerous than unchanged lymphoma after initial therapy.

Whereas the clinical characteristics of our collective appear representative for gastric MALT lymphoma when compared to other recent series, the overall response rate to upfront antibiotics in our unselected series appeared to be lower than expected, particularly for the H. pylori-positive cohort (overall response rate 62%); the relatively large cohort of H. pylori-negative patients had a good response rate at 46% but no further conclusions in terms of clinical characteristics or PFS of H. pylori-negative patients responding to eradication could be drawn due to the low number of patients in this group (*n* = 22, responders *n* = 10). In addition, and with the caveat of the small number of patients treated with upfront systemic therapy, i.e., chemo- and/ or immunotherapy (32/137, 23%), documented response rates in this cohort were excellent (see Table 2). Still, upfront systemic treatment for gastric MALT lymphoma should be restricted to patients with symptomatic H. pylori-negative or disseminated disease.

A potential weakness of this analysis is that we did not correlate data with translocation t(11; 18) (q21; q21) status, which has been associated with gastric MALT lymphoma not responding to eradication treatment; however, according to the current guidelines, the routine analysis of t(11; 18) is not recommended in the upfront diagnosis of patients with gastric MALT lymphoma [6,7]. In view of this, we stopped analyzing this after the recommendations from the EGILS (European Gastro-Intestinal Lymphoma Study) cohort in 2011 were published and did that mostly on an individual basis or if systemic therapies outside of clinical trials were further planned. Thus, we cannot offer a representative unbiased cohort of patients in order to answer that question.

## 5. Conclusions

To conclude, while our data underline the favorable prognosis of gastric MALT lymphoma in general with a 5-year survival > 90%, these results underscore the assumption that depth of remission is a relevant prognostic marker for PFS following first anti-lymphoma treatment irrespective of the applied treatment modality. While our data do not justify an aggressive approach with early oncological intervention to force a complete remission, they suggest the development of more risk-stratified follow-up algorithms, potentially reducing the number of endoscopic controls for the individual patient in the absence of increased risk for progression, transformation or death.

## Figures and Tables

**Figure 1 cancers-12-00492-f001:**
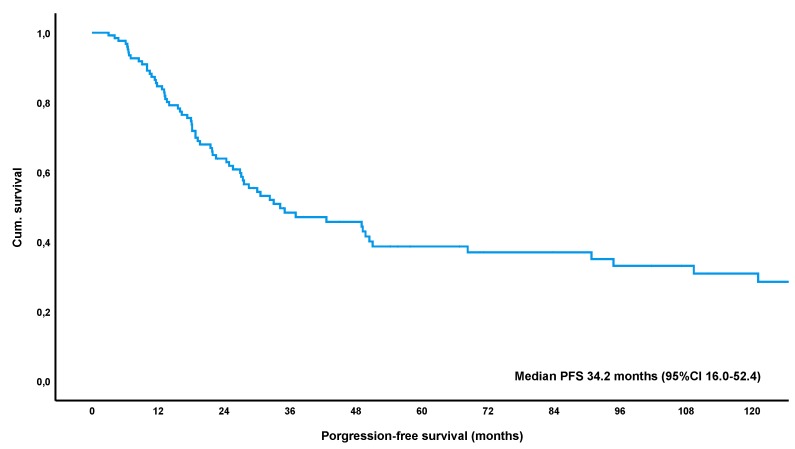
Kaplan-Meier curve for progression-free survival following first-line treatment for patients with gastric mucosa-associated lymphoid tissue (MALT) lymphoma. Abbreviations: CI = confidence interval.

**Figure 2 cancers-12-00492-f002:**
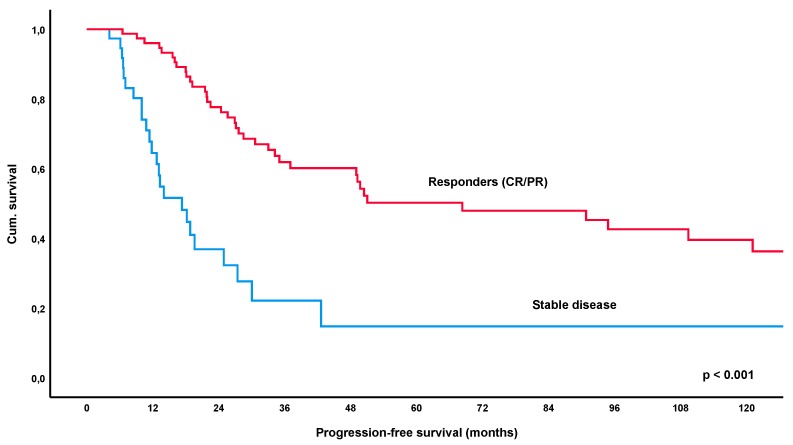
Kaplan-Meier curve for progression-free survival following first-line treatment (various modalities), comparing patients with an objective response (=partial or complete remission) versus stable disease. Abbreviations: CR = complete remission, PR = partial remission.

**Figure 3 cancers-12-00492-f003:**
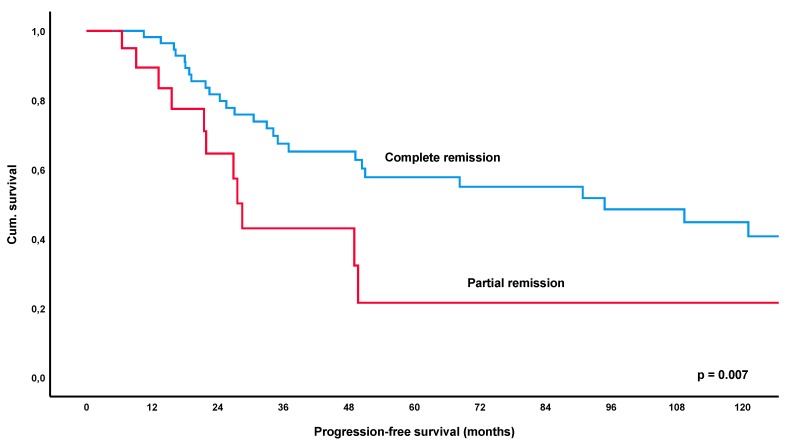
Kaplan-Meier curve for progression-free survival following first-line treatment (various modalities), comparing patients with complete remission versus patients achieving partial remission.

**Figure 4 cancers-12-00492-f004:**
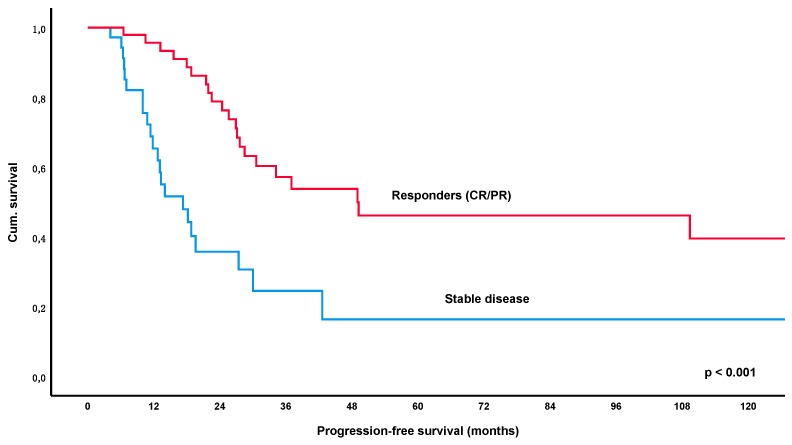
Kaplan-Meier curve for progression-free survival following first line *Helicobacter pylori* eradication, comparing patients with an objective response (=partial or complete remission) versus stable disease. Abbreviations: CR = complete remission, PR = partial remission.

**Figure 5 cancers-12-00492-f005:**
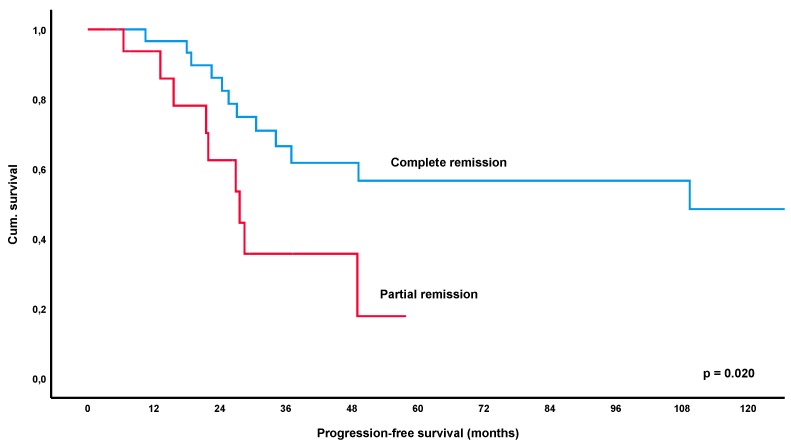
Kaplan-Meier curve for progression-free survival following first *Helicobacter pylori* eradication, comparing patients with complete remission versus patients achieving partial remission.

**Table 1 cancers-12-00492-t001:** Baseline characteristics gastric mucosa-associated lymphoid tissue (MALT) lymphoma patients (*n* = 137 patients).

Parameter	Number of Patients
Sex (Female/Male)	51% (70/137)/49% (67/137)
Median age (range)	63 years (22–85 years)
Age ≥ 70 years	28% (38/137)
Stage of disease-Ann Arbor	
Ann Arbor IE	66% (90/137)
Ann Arbor IIE	23% (32/137)
Ann Arbor IIIE	1% (1/137)
Ann Arbor IV	10% (14/137)
Stage of disease-Lugano	
Lugano I	66% (90/137)
Lugano II	23% (32/137)
Lugano IV	11% (15/137)
MALT-IPI	
Low risk	62% (75/122)
Intermediate risk	33% (40/122)
High risk	6% (7/122)
Further clinical features	
Helicobacter pylori positive	68% (93/137)
Plasmacellular differentiation	22% (22/101)
LDH > upper normal limit	7% (8/122)
Autoimmune disorder	20% (24/120)
Beta-2-micorglobuline > UNL	15% (19/130)
Hepatitis B/C virus	2% (2/121)
Paraproteinemia	27% (25/91)
Median follow-up time (interquartile range)	56.2 months (26.3–111.6)

MALT-IPI = MALT lymphoma prognostic index; LDH = lactate dehydrogenase; UNL = upper normal limit.

**Table 2 cancers-12-00492-t002:** First-line treatment characteristics and response data in gastric MALT lymphoma patients (*n* = 137).

Treatment/Response	Number of Patients
**First-line treatment overall collective**	
*Helicobacter pylori* eradication	70% (96/137)
Systemic treatment (chemo-/immunotherapy)	23% (32/137)
Local therapy (radiation, surgery)	4% (5/137)
Watch and wait	3% (4/137)
**First-line treatment *Helicobacter pylori* negative**	
*Helicobacter pylori* eradication	50% (22/44)
Systemic treatment (chemo-/immunotherapy)	41% (18/44)
Local therapy (radiation, surgery)	5% (2/44)
Watch and wait	5% (2/44)
**Response to treatment overall collective**	
Overall response rate	67% (86/128)
Complete remission	48% (62/128)
Partial remission	19% (24/128)
Stable disease	30% (38/128)
Progressive disease	3% (4/128)
**Response to eradication therapy**	
Overall response rate	58% (54/93)
Complete remission	38% (35/93)
Partial remission	20% (19/93)
Stable disease	39% (36/93)
Progressive disease	3% (3/93)
**Response to systemic treatment ***	
Overall response rate	90% (27/30)
Complete remission	77% (23/30)
Partial remission	13% (4/30)
Stable disease	7% (2/30)
Progressive disease	3% (1/30)
**Response to local treatment**	
Overall response rate	100% (5/5)
Complete remission	80% (4/5)
Partial remission	20% (1/5)

* systemic treatment includes: chemotherapy ± rituximab (*n* = 23), immunomodulatory compounds (*n* = 4), anti-CD20-antibody monotherapy (*n* = 3). The text in bold indicates the distinct group presented.

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
