# Peer review of "Depth of Remission Following First-Line Treatment Is an Independent Prognostic Marker for Progression-Free Survival in Gastric Mucosa-Associated Lymphoid Tissue (MALT) Lymphoma"

_cancers, 2020, doi:10.3390/cancers12020492_

Round 1

Reviewer 1 Report

Kiesewetter and colleagues examine outcomes for patients with gastric MALT lymphomas treated over a 20 year period at their institution. Not surprisingly, strength of response to initial therapy correlated with progression free survival time, both in the group as a whole and for h. pylori positive patients treated with antibiotic erradication therapy alone. There were no differences in overall survival. Results are confirmatory of clinical studies, but these results are of interest to readers given the large sample size and quality of analysis. These data could influence clinical decisions at the level of frequency of followup endoscopic procedures needed for patients based on initial responses to therapy.

An important factor excluded from analysis, however, was presence of the recurrent t(11;18) chromosomal abnormality that correlates with more advanced disease at presentation and worse response to h. pylori eradication. This known risk factor is not analyzed or discussed, and this should be included to the extent possible in a minor revision. It would be very interesting to know the effect of this known disease driver in this relatively large cohort.

Author Response

Response to Reviewer 1:
Thank you for your review of our paper. Please find here a reply to your comment:

1.) An important factor excluded from analysis, however, was presence of the recurrent t(11;18) chromosomal abnormality that correlates with more advanced disease at presentation and worse response to h. pylori eradication. This known risk factor is not analyzed or discussed, and this should be included to the extent possible in a minor revision. It would be very interesting to know the effect of this known disease driver in this relatively large cohort.
Reply: Thank you for the feedback and the very important comment.
We understand, that a potential weakness of our analysis is that we did not correlate data with translocation t(11;18)(q21;q21) status, which has been associated with gastric MALT lymphoma not responding to eradication treatment, however, according to the current guidelines, the routine analysis of t(11;18) is not recommended in the upfront diagnosis of patients with gastric MALT lymphoma. In view of this, we have stopped analyzing this after the recommendations from the EGILS cohort in 2011 were published (Ruskoné-Fourmestraux A et al, Gut 2011); and have done that mostly on an individual basis or if systemic therapies outside of clinical trials were further planned. Thus, we cannot offer a representative unbiased cohort of patients in order to answer that question, as it includes a much higher rate of positive patients than expected in an unselected cohort.

We have now added a corresponding statement to the discussion section of our paper (last paragraph page 11). In addition we would like to explain, that the objective of this trial was not to assess factors predicting response (that has already been published), but to assess the influence of depth of remission on PFS and outcome. The reviewer also states that it might be of interest, should the data be available, but as it is formulated, it does not appear a prerequisite for the general judgement of our paper. Re-assessment of t(11;18) in all our patients included in this analysis, however, would unfortunately not be possible within the time-frame given for the revision.

Reviewer 2 Report

This is a well-written paper on a fairly large series of gastric MALT lymphoma with adequate follow-up. The authors show that depth of remission is a good marker for progression free survival, including cases with only PR, and irrespective of Helicobacter status and firstline treatment. The data confirm previous studies, but provide insight into the outcome of patients with PR, thus giving more information on clinical handling. 

Specific points:

The authors have collected a good amount of data which are not further mentioned in the results section, including presence or absence of paraprotein. and plasmacytic differentiation. One can assume that these parameters did not influence PFS, but it should be mentioned briefly.

It is well known that response to H. pylori eradication is associated with absence of the t(11;18) translocation. It would be interesting to know whether this was also the case in this series, if the data are available. 

Did the authors observe any differences between the H.p. negative cases which did respond to eradication therapy versus those which did not?

Author Response

Response to Reviewer 2:

Thank you for your review of our paper. Please find here a point-by-point reply to your comments:

1.) The authors have collected a good amount of data which are not further mentioned in the results section, including presence or absence of paraproteins and plasmacytic differentiation. One can assume that these parameters did not influence PFS, but it should be mentioned briefly.
Reply: Thank you for the feedback. “No correlation between PFS and diagnosis of autoimmune disorder, plasmacellular differentiation or presence of paraproteinemia was observed (p-values all non-significant).” This statement was added for both the entire cohort (results page 6) and the H. pylori eradication only cohort (results page 8), respectively.

2.) It is well known that response to H. pylori eradication is associated with absence of the t(11;18) translocation. It would be interesting to know whether this was also the case in this series, if the data are available.

Reply: Thank you for this very important comment.
We understand, that a potential weakness of our analysis is that we did not correlate data with translocation t(11;18)(q21;q21) status, which has been associated with gastric MALT lymphoma not responding to eradication treatment, however, according to the current guidelines, the routine analysis of t(11;18) is not recommended in the upfront diagnosis of patients with gastric MALT lymphoma. In view of this, we have stopped analyzing this after the recommendations from the EGILS cohort in 2011 were published (Ruskoné-Fourmestraux A et al, Gut 2011); and have done that mostly on an individual basis or if systemic therapies outside of clinical trials were further planned. Thus, we cannot offer a representative unbiased cohort of patients in order to answer that question, as it includes a much higher rate of positive patients than expected in an unselected cohort. We have now added a corresponding statement to the discussion section of our paper (last paragraph page 11). In addition we would like to explain, that the objective of this trial was not to assess factors predicting response (that has already been published), but to assess the influence of depth of remission on PFS and outcome. The reviewer also states that it might be of interest, should the data be available, but as it is formulated, it does not appear a prerequisite for the
[email protected]
general judgement of our paper. Re-assessment of t(11;18) in all our patients included in this analysis, however, would unfortunately not be possible within the time-frame given for the revision.

3.) Did the authors observe any differences between the H.p. negative cases which did respond to eradication therapy versus those which did not?

Reply: Response to eradication treatment in H. pylori negative patients was not significantly influenced by clinical characteristics i.e. presence of autoimmune disorder, plasmacytic differentiation, paraproteinemia or MALT-IPI factors. As these data are however difficult to interpret with regard to the low numbers in this specific group (responders n = 10, total number of patients n = 22), we did not include this in the results section but added a corresponding statement to the discussion (page 11): “Whereas the clinical characteristics of our collective appear representative for gastric MALT lymphoma when compared to other recent series, the overall response rate to upfront antibiotics in our unselected series appeared to be lower than expected, particularly for the H. pylori positive cohort (overall response rate 62%); the relatively large cohort of H. pylori negative patients had a good response rate at 46% but no further conclusions in terms of clinical characteristics or PFS of H. pylori negative patients responding to eradication could be drawn due to the low number of patients in this group (n = 22, responders n = 10).”